# Conductive Mediators in Oxidation Based on Ferrocene Functionalized Phosphonium Ionic Liquids

**DOI:** 10.3390/ijms232415534

**Published:** 2022-12-08

**Authors:** Vadim V. Ermolaev, Liliya R. Kadyrgulova, Mikhail N. Khrizanforov, Tatiana P. Gerasimova, Gulnaz R. Baembitova, Anna A. Lazareva, Vasili A. Miluykov

**Affiliations:** 1Arbuzov Institute of Organic and Physical Chemistry, FRC Kazan Scientific Center, Russian Academy of Sciences, 8 Arbuzov Street, 420088 Kazan, Russia; 2A.M. Butlerov Chemistry Institute, Kazan Federal University, Kremlevskaya Str. 18, 420008 Kazan, Russia

**Keywords:** phosphonium salts, mediators, ferrocene, ionic liquid, synthesis, electrochemical properties

## Abstract

Herein, the synthesis of ferrocene-containing salts is presented. Acylation of ferrocene (Fc) according to the Friedel–Crafts method led to ω-bromoacyl ferrocenes. The ω-bromoacyl ferrocenes were subsequently introduced to quaternization reaction with tri-*tert*-butyl phosphine, which resulted in phosphonium salts. Obtained phosphonium salts were characterized by physical methods. The electrochemical properties of phosphonium salts were studied by cyclic voltammetry (CV). It was found that the replacement of *n*-butyl fragments at the phosphorus atom by *tert*-butyl leads to a more anodic potential shift. In contrast to isolobal structures Fc-C(O)(CH_2_)_n_P^+^(*n*-Bu)_3_X^−^ and Fc-(CH_2_)_n+1_P^+^(*n*-Bu)_3_X^−^, the CV curves of Fc-C(O)(CH_2_)_n_P^+^(*t*-Bu)_3_X^−^ and Fc-(CH_2_)_n+1_P^+^(*t*-Bu)_3_X^−^ did not show a large discrepancy between forward and reverse currents. The transformation of the C=O groups to CH_2_ fragments had a significant effect on the electrochemical properties of ferrocene salts, the oxidation potential of which is close to that of pure ferrocene.

## 1. Introduction

In recent years, a significant number of reports has been devoted to materials that respond to external influences [1]. These materials find applications in the most advanced fields, such as sensors, electronic and optical technologies, biomedicine, molecular machines, and biomimetics [2]. Ionic liquids (ILs) are attractive components for creating such materials due to the fact that their chemical and physical properties can be easily tuned according to the desired application by carefully selecting cations and anions [3,4]. This certainly makes them excellent candidates as building blocks for “smart” materials [5]. A comparison of ammonium and phosphonium ionic liquids shows the advantages of the latter, since such compounds have higher thermal and chemical stability, as well as increased biological activity [6,7,8].

At the same time, the addition of different individual molecular synthons to ionic liquids can bring new desired properties. For example, it was shown that ferrocene-containing ionic liquids demonstrated important electrochemical, magnetic, and other properties of ILs [9,10,11,12].

Despite the fact that the ferrocene was discovered almost seventy years ago, the interest of chemists has not faded away [13]. Ferrocene itself and ferrocene-containing compounds find wide application [14]. Thus, it has been shown that ferrocenyl-substituted compounds, such as ferroceneacetic acid and ferrocenemethanol, can act as mediators of ascorbic acid oxidation, i.e., reducing the reaction potential [15].

The unique electrochemical properties of ferrocene allow for the design of electrochromic devices with enhanced properties. A fast response, high contrast, and lower voltage were achieved for the composition of viologen and redox-active ionic liquid containing a ferrocenyl fragment [16,17]. This combined material does not require any additive that is commonly used in the creation of electrochromic devices, such as solvents, supporting electrolytes, and sacrificial agents. On the other hand, the ferrocene fragment can influence the biological activity. Increased activity against *Escherichia coli* and *Staphylococcus aureus* bacteria has been demonstrated in imidazolium ionic liquids and IL membranes. This is explained by the formation of active oxygen particles caused by the presence of the redox-active Fc-moiety [18]. The universality of ferrocene chemistry is proved by using core-shell magnetic nanoparticles decorated with ferrocene-containing ILs as a nickel catalyst support in the synthesis of various pyrans’ derivatives in a one-pot manner under ultrasonic conditions [19,20]. A large amount of work was completed by Garib and Hirsch. A series of several new families of ferrocene-containing imidazolium salts were synthesized by etherification of ferrocene methanol, acylation of ferrocene, and amide coupling of ferrocene carboxylic acid [21]. These new redox-active ionic liquids were fully characterized.

At the same time, Fc-containing phosphonium ionic liquids are presented much less often in the literature. One of the first publications devoted to the electrochemical properties of ferrocene-functionalized phosphonium IL was reported in 2011 [22]. Ferrocene fragment was linked to tri-*n*-butyl phosphine moiety through C8 alkyl or acyl spacer. The authors discovered that Fc^0/+^ self-exchange reactions were responsible for electron transfer as opposed to physical diffusion of the phosphonium species. Later, Sundermeyer described phosphonium salt where the Fc-fragment was connected to the P-atom directly [23]. Several routes to achieve the desired product were evaluated.

We demonstrated that steric hindrance at phosphorus atom could play a key role in different aspects of phosphonium ionic liquids’ application [24,25,26]. Due to unique physical properties such as high conductivity, a very wide electrochemical window, stability, and reproducibility, phosphonium IL with sterically hindered cation was successfully applied in an electrochemical study of insoluble compounds [27,28]. Recently, we reported the synthesis and physical properties of the series of the ferrocenyl-containing sterically hindered phosphonium salts based on di(*tert*-butyl)ferrocenyl phosphine [29,30]. In development of this concept, we extended the series of ferrocene-containing phosphonium salts, where the ferrocene fragment and the phosphorus atom are linked by a spacer of different length and nature. Analysis of voltammograms of the obtained compounds revealed correlations between their structures and electrochemical properties.

## 2. Results and Discussion

In this study, the synthesis of ferrocene acyl phosphonium salts is presented. In the first step, ω-bromoacyl chlorides were obtained from corresponding carboxylic acid and thionyl chloride. The product was isolated by distillation under reduced pressure and then introduced in the reaction with ferrocene in the presence of Lewis acid (Figure 1).

This synthesis was performed according to Hirsch’s report; however, the synthesis was performed without Zn powder. This allows obtaining the single desired product that was purified by column chromatography on silica.

At the next stage, the quaternization reaction of compounds **2a**–**d** with tri-*tert*-butyl-phosphine was carried out under inert atmosphere in absolute MeCN at the boiling point of MeCN. The reaction was monitored by ^31^P NMR. After the reaction, the finished product was washed several times with petroleum ether and diethyl ether (Figure 2).

For further electrochemical study, the bromide-ion was changed to BF_4_^−^ by simple anion exchange reaction in a minimal amount of water with subsequent extraction of salt with CH_2_Cl_2_. The organic layer was collected and washed several times with water and dried over Na_2_SO_4_. After filtration, the solvent was removed in vacuum, resulting in dark orange-brown-colored honey-like amorphous substances.

The obtained salts were characterized with the NMR ^1^H, ^13^C, ^31^P (Table 1). In the phosphorus spectrum, only one singlet at 49–50 ppm characteristic for tri-*tert*-butyl phosphonium salts [31] was observed.

At the same time, anion exchange leads to changes in the chemical shifts of protons P–CH_2_. They are not of the same nature due to the fact that they are exposed to different factors (location of the ferrocene fragment, carbonyl group, and phosphorus atom). The move of the chemical shift to high fields when Br^−^ was replaced with BF_4_^−^ in the case of salts **3a**–**4a** can be explained by the proximity of the carbonyl group. The same movement of the chemical shift for **3c**–**4c** is possibly connected with interaction of carbonyl group trough space. Pairs **3b**–**4b** and **3d**–**4d** behave in the same was as like salts, with a normal chain as the fourth substituent.

Thermo-gravimetric analysis of obtained salts was performed (see Appendix A). Only one salt, **4a,** possesses phase transformation at 167 °C (Figure 1). This is probably connected with the fact that the salt with the short spacer forms some sort of ordered structure. All other salts did not demonstrate phase transition due to its honey-like amorphous state. It was found that all tetrafluoroborates are more thermally stable (up to 200 °C) in contrast to their bromides’ analogues (about 150 °C).

After thermal decomposition began, jitter occurred on all DSC graphs. Most likely it is caused by the vibration of the sample holder lid induced by intense exhaust gases.

It was interesting to obtain phosphonium salts containing an alkyl chain as a linker. To study the mutual influence of the carbonyl moiety on the cyclopentadienyl and phosphonium moieties, we carried out a reduction reaction using *tert*-butylamine-borane in the presence of Lewis acid (Figure 3).

The reaction mixture was treated with water and products were extracted with CH_2_Cl_2_. Concentrated solution was placed in chromatographic column and the isolated ferrocenyl bromoalkanes were introduced in a quaternization reaction with tri-*tret*-butyl phosphine (Figure 4). Anion replacement was carried out by interaction of the halides with a twofold excess of sodium tetrafluoroborate or bis(trifluoromethanesulfonyl)imide lithium. By employing one of such salts, we carried out a reversible electrochemical change in the color of the solution.

The electrochemical properties of phosphonium salts **4a**–**d** and **8a**–**d** were studied by cyclic voltammetry and compiled with data from the literature analogue (Figure 2; Table 1). It was found that the replacement of *n*-Bu fragments at the phosphorus atom with *t*-Bu led to a more anodic potential shift (Table 2).

The observed effect can be described by the fact that, in addition to the acceptor properties of the acyl group, on the electronic effect in the case of salts with *t*-Bu, phosphorus, due to the spatial effect, pulls away a part of the electron density of the cyclopentadiene ring.

R.W. Murray et al. [22] also noted that the Fc-C(O)(CH_2_)_n_P(*n*-Bu)_3_X^−^ and Fc-C(CH_2_)_n+1_P(*n*-Bu)_3_X^−^ salts have weak ionic conductivity. On the CV curves, a large discrepancy between the forward and reverse currents was observed. Such an effect is not observed for salts **4a**–**d** and **8a**–**d**. On the contrary, these salts can be used as an electrolyte.

It was noted earlier that, based on previous works, the phosphonium cation with normal alkyl substituents has a low diffusion coefficient (10^−8^–10^−9^ cm^2^/s), which is associated with the interaction of the chains with the medium. At the same time, the sterically hindered phosphonium cation described in this work is more compact, which is reflected in the high *D* value (Table 2).

When iron is oxidized (to ferrocenium form), a characteristic color change from yellow to dark green is observed; when re-reduced, the system reversibly passes into its original form (Appendix A). The color change is controlled by the difference between forward and reverse current.

**Table 2 ijms-23-15534-t002:** Electrochemical data for the redox properties of **4a**–**d** and **8a**–**d**.

Compound	^1/2^E_ox_, V vs. Ag/AgCl	ΔE_PEAK,_ V	*D*_CV_, cm^2^/s
Ferrocene	0.41	0.06	1.30 × 10^−7^ [32]
**4a**	0.77	0.06	1.72 × 10^−5^
**4b**	0.71	0.06	1.71 × 10^−5^
**4c**	0.71	0.06	1.71 × 10^−5^
The analog of **4c** (Fc-C(O)(CH_2_)_5_P(*n*-Bu)_3_PF_6_^−^) [22]	0.60	0.30	6.40 × 10^−9^
**4d**	0.71	0.06	1.68 × 10^−5^
**8a**	0.46	0.06	1.64 × 10^−5^
**8b**	0.44	0.06	1.65 × 10^−5^
**8c**	0.43	0.06	1.62 × 10^−5^
The analog of **8c** (Fc(CH_2_)_6_P(*n*-Bu)_3_PF_6_^−^) [22]	0.25	0.30	1.53 × 10^−8^
**8d**	0.42	0.06	1.58 × 10^−5^

The diffusion coefficients *D***_CV_** were calculated based on the electrochemical data [33].

At CVs for all salts, the ΔE_PEAK_ does not exceed 60 mV. The transformation of the C=O groups to CH_2_ fragments has a significant effect on the electrochemical properties of ferrocene salts, the oxidation potential of which is close to that of pure ferrocene. An increase in the alkyl chain in the composition of ferrocene salts does not lead to a significant potential shift (when going from C3 to C11, the potential shift remains 50 mV) (averaged measurements of 10 experiments for each substance).

The generated ferrocenium can be used as an oxidizing agent (or oxidation mediator) for, for example, ascorbic acid. When ascorbic acid is added to a solution of oxidized ferrocenyl-phosphonium salt, a change in the color of the solution to the original yellow color is observed, which characterizes the implementation of the oxidation reaction (Figure 3).

The color changes can be detected by UV/Vis spectroscopy. The initial yellow salt **7a** absorbs at 434 nm (black spectrum at Figure 4) that corresponds to the Fc-moiety. After oxidation, an additional band at 463 nm appears (green spectrum at Figure 4). The addition of ascorbic acid to neutral **7a** leads to the appearance of its own band at 241 nm (red spectrum at Figure 4). However, in the spectrum of reaction mixture, the intensity of this band significantly decreases due to its oxidation (blue spectrum at Figure 4), whereas the spectral pattern at 350–550 nm is close to the observed for oxidized form of **7a**.

This, in turn, demonstrates rather mild oxidation conditions, which makes ferrocenyl-phosphonium salts convenient electrolytes and mediators at the same time. On the CV curves (Figure 5), with the addition of increasing amounts of ascorbic acid, an increase in current is observed corresponding to the transition of Fe (II) to Fe (III).

The reaction mixture and initial compounds were analyzed by ESI-MS. In the spectrum of the reaction mixture, we did not observe any peak of the starting ascorbic acid. Signals of oxidative degradation products are recorded, which are comparable to the literature data [34]. According to the spectrum, we observed products of both partial and complete oxidative degradation (Appendix A).

Thus, given the high ionic conductivity, the obtained salts can be used as oxidation mediators, where the use of additional electrolytes is not required.

## 3. Materials and Methods

### 3.1. NMR Experiments

NMR spectra were recorded with multi-nuclear spectrometer Bruker AVANCE-400 (BRUKER BioSpin GMBH am Silberstreifen, D-76287, Rheinstetten, Germany) (400.1 MHz (^1^H), 100.6 MHz (^13^C) and 162.0 MHz (^31^P)). Chemical shifts were given in parts per million relative to SiMe_4_ (^1^H, internal solvent) and 85% H_3_PO_4_ (^31^P, external).

### 3.2. TG-DSC

Thermogravimetric analysis was performed on the NETZSCH STA 449F1 (NETZSCH, Selb, Germany) with a heating rate 10 K per minute up to 400 °C in an argon atmosphere.

### 3.3. Mass-Spectra

Mass Electrospray ionization mass spectrometry (ESI-MS) was performed on the AmazonX mass spectrometer (Bruker Daltonik GmbH, Bremen, Germany). The measurements were carried out in the positive/negative ion detection mode in the *m*/*z* range from 100 to 1000. The voltage on the capillary was 140 V. Data were processed using the DataAnalysis 4.0 program (Bruker Daltonik GmbH, Bremen, Germany).

### 3.4. Electrochemical Measurements

Electrochemical measurements were taken on a BASi Epsilon EClipse electrochemical analyzer (West Lafayette, IN, USA). The program concerned Epsilon-ECUSB-V200 waves at potential scan rate t = 100 mV∙s^−1^ in CH_3_CN/without an electrolyte (except clear ascorbic acid—for ascorbic acid, a 0.1 M solution of Bu_4_NBF_4_ was used) at 295 K. A glassy carbon working electrode (ð = 3 mm) embedded in Teflon and Pt wire as counter electrode was used in the electrochemical cell. Before each measurement, the surface of the working electrode was mechanically polished.

### 3.5. Reagents and Research Subjects

All the work related to the preparation of the starting substrates, as well as the synthesis and the workup of products, was carried out in an inert atmosphere using the standard Schlenk apparatus. Tri-*tert*-butylphosphine (Dal-Chem, Nizhny Novgorod, Russia), halogenated acids (Sigma-Aldrich, St. Louis, MO, USA), anhydrous aluminum chloride (Alfa Aesar, Heysham, Lancashire, UK), Thionyl Chlride (Himreaktiv, Nizhny Novgorod, Russia), Ferrocene (Thermo Fisher (Kandel) GmbH, Kandel, Germany). All solvents and purchased reagents were absolute by the appropriate methods, mainly by distillation in an inert atmosphere. Detailed characterization of the phosphonium salts described in Appendix A.

### 3.6. General Procedure

#### 3.6.1. General Procedure for the Synthesis of ω-Bromoalkanoyl Chloride **1a**–**1d** (Scheme S1)

At room temperature, halogenated acid (1 eq.) and SOCl_2_ (4.2 eq.) were added with a stirrer to a 50 mL Schlenk vessel and gradually warmed to reflux. The apparatus should be connected to a bubbler and an exhaust gas absorbing device. The reaction mixture was refluxed within 4 h and then was stirred within 12 h at room temperature. The crude product was purified by vacuum distillation.

#### 3.6.2. General Procedure for the Synthesis of **2a**–**2d**

Under an inert gas atmosphere, the ω-bromoalkanoyl chloride (1 eq.) and anhydrous AlCl_3_ (1 eq.) were added to dry CH_2_Cl_2_ (50 mL) at 0 °C. The mixture was stirred for ten minutes before the addition of ferrocene (1.2 eq.) at the same temperature. The appearance of a purple color signaled the beginning of the acylation process. Stirring was continued after addition overnight while the mixture was allowed to warm up to room temperature. The mixture was again cooled to 0 °C by an ice bath, and water (40 mL) was carefully added under an inert gas atmosphere to quench the reaction. After quenching, more H_2_O (50 mL) was added and the mixture was stirred for 30 min at 0 °C. The aqueous phase was extracted four times with CH_2_Cl_2_ (4 × 60 mL). The combined organic extracts were washed sequentially with sat. NaHCO_3_ (2 × 100 mL), H_2_O (100 mL), and brine (100 mL). The organic phase was dried over MgSO_4_, concentrated, and the residue purified by column chromatography (SiO_2_; petroleum ether/ethyl acetate). All products were dried in vacuo after purification.

#### 3.6.3. General Procedure for the Synthesis of **3a**–**3d**

Bromoalkanoyl ferrocene was dissolved in dry MeCN (20 mL) and an equivalent amount of tri-*tert*-butyl phosphine was added. The reaction mixture was stirred at 82 °C for 6 h. The solvent was removed in vacuo and the product was stirred with 3 portions of 20 mL of petroleum ether and 3 portions of 20 mL of diethyl ether. After each stirring, the solvent was removed with a cannula with a filter nozzle, and the solvent residues were removed in vacuo.

#### 3.6.4. General Procedure for the Synthesis of **4a**–**4d**

Salts **3a**–**d** were dissolved in 10 mL of water, and a twofold excess of sodium tetrafluoroborate solution in water was added to them. The reaction mixture was stirred for 12 h. The precipitate was filtered, and the solvent residues were removed in vacuo. The salt was dissolved in 20 mL of methylene chloride and washed with distilled water. The organic phase was separated and dried over magnesium sulfate. The solvent was evaporated, and the salt was dried under vacuum at 40 °C for 8 h.

#### 3.6.5. General Procedure for the Synthesis of **5a**–**5d**

Salts **3a**–**d** were dissolved in 10 mL of water, a one and a half excess of lithium bis(trifluoromethanesulfonyl)imide solution in water was added to them. The reaction mixture was stirred for 1 h. The product was extracted with CH_2_Cl_2_. The extract was washed with distilled water. The organic phase was separated and dried over magnesium sulfate. The solvent was evaporated, and the salt was dried under vacuum at 50 °C for 8 h.

#### 3.6.6. General Procedure for the Synthesis of **6a**–**6d**

The *tert*-Butylamine borane (6 eq.) was added to a suspension of anhydrous AlCl_3_ (3 eq.) in dry CH_2_Cl_2_ (40 mL) at 0 °C under an inert gas atmosphere. The resulting mixture was stirred at the same temperature for 1 h. The solution of bromoalkanoyl ferrocene (1 eq.) in CH_2_Cl_2_ was added dropwise. The resulting solution was stirred overnight. The initial purple color of the solution changed to brown during the reduction process. This indicated the end of the reaction. The reaction mixture was cooled to 0 °C and water (20 mL) was added dropwise to remove excess of the reducing agents. More H_2_O (20 mL) was added and the solution was stirred for 30 min. The phases were separated and the aqueous phase was extracted with CH_2_Cl_2_ until the organic extract was colorless. The combined organic extracts were sequentially washed with sat. NaHCO_3_ (2 × 20 mL), H_2_O (20 mL), and brine (20 mL). The organic phase was dried over MgSO_4,_ and all products were dried in vacuo after purification.

#### 3.6.7. General Procedure for the Synthesis of **7a**–**7d**

Tri-*tert*-butyl phosphine was dissolved in dry DMF (5 mL) and an equivalent amount of bromoalkyl ferrocene was added. The reaction mixture was stirred at 55 °C for 5–20 h. The solvent was removed in vacuo and the product was stirred with 3 portions of 20 mL of petroleum ether and 3 portions of 20 mL of diethyl ether. The precipitate was filtered out and the solvent residues were removed in vacuo.

#### 3.6.8. General Procedure for the Synthesis of **8a**–**8d**

Salts **6a**–**d** were dissolved in 10 mL of EtOH, and a one and a half excess of lithium bis(trifluoromethanesulfonyl)imide solution in EtOH was added to them. The reaction mixture was stirred for 1 h. The product was extracted with CH_2_Cl_2_. The extract was washed with distilled water. The organic phase was separated and dried over magnesium sulfate. The solvent was evaporated, and the salt was dried under vacuum at 50 °C for 8 h.

## 4. Conclusions

In this study, the synthesis of ferrocene containing phosphonium salts with sterically hindered cation was described. Obtained salts were characterized with several physical methods which are listed as follows: NMR ^1^H; ^13^C; ^31^P; ESI-MS; TG-DSC; IR.

It was found by cyclic voltammetry that the replacement of *n*-Bu fragments at the phosphorus atom with *t*-Bu leads to a more anodic potential shift. For the obtained phosphonium salts on CV, the discrepancy between forward and reverse currents does not exceed 60 mV, in contrast to isolobal structures with *n*-alkyl fragments. It was established that the phosphonium cation with *t*-Bu substituents has a high diffusion coefficient, which is associated with a more compact structure. In this regard, salts can be simultaneously used as mediators and electrolytes.

We suppose that the ferrocene moiety is responsible for the cation stabilization and facilitating the charge transfer. Altogether, this helps to define the trends in fundamental research on new materials for electrochemical application.

## Data Availability

The data presented in this study are contained within the article or are available upon request from the corresponding author, Mikhail N. Khrizanforov.

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
