# Peer review of "Conductive Mediators in Oxidation Based on Ferrocene Functionalized Phosphonium Ionic Liquids"

_ijms, 2022, doi:10.3390/ijms232415534_

Round 1

Reviewer 1 Report

The manuscript ijms-2059035 entitled “Conductive mediators in oxidation based on ferrocene functionalized phosphonium ionic liquids” was reviewed carefully. The manuscript is clear, and concise and the work is original. So, it needs major revision, due to some of the specific comments which are listed below:

1.         Please improve English writing.

2.         Electrochemical investigations are incomplete and have some errors:

2.1.      Electrochemical behavior investigation can be completed more by calculating the diffusion coefficient or alpha.

2.2.      The data in Figure 2 do not match the data in Table 2.

2.3.      Line 160: The meaning of the sentence “The color change is controlled by the difference between forward and reverse current.” is unclear. Voltammetric currents are in the range of microamperes and do not lead to color change!

2.4.      The reference of the article data used in Table 2 should be specified in the table.

2.5.      The electrochemical behavior of the Ferrocene/Ferrocenium couple is completely reversible, that is, the peak current depends on the total concentration of these two, not on the concentration of each separately. As a result, the argument presented for Figure 5 is completely wrong (Lines 190 and 191).

3.         Line 136: Scheme 6 is referenced, while there is no Scheme 6 in the main article and supplementary Information file.

4.         The caption of scheme 4 is not complete, write it in a complete form.

5.         The title of table 2 is wrong, please correct it.

Author Response

Dear Reviewer,

Herewith we submit a revised manuscript by Vadim V. Ermolaev, Liliya R. Kadyrgulova, Mikhail N. Khrizanforov*, Tatiana P. Gerasimova, Gulnaz R. Baembitova, Anna A. Lazareva, Vasili A. Miluykov «Conductive mediators in oxidation based on ferrocene func-tionalized phosphonium ionic liquids» to the International Journal of Molecular Sciences for participation in the thematic issue « Recent Advances in Novel Compositions for Electrochemical Applications)»

First of all, we would like to thank the Reviewers for inspective reading and reviewing of our manuscript and their valuable remarks.

We are confident that our results provide new important insights into the organic, organophosphorus and organometallic chemistry and such would be of a substantial interest to the broad interdisciplinary readership of the “International Journal of Molecular Sciences”. We therefore strongly hope that our work can be accepted for publication in your Journal.

We also provide the detailed answers to the reviewers’ remarks.

Please, do not hesitate to contact me if you have any problems or questions regarding our manuscript or if you have difficulty opening the files.

Yours sincerely,

Dr. Mikhail Khrizanforov,

======================================

Response to Reviewer 1 Comments

The manuscript ijms-2059035 entitled “Conductive mediators in oxidation based on ferrocene functionalized phosphonium ionic liquids” was ‎reviewed ‎carefully. The manuscript is clear, and concise and the work is original. So, it needs major ‎revision, ‎due to some of ‎the ‎specific ‎comments which are listed ‎below:

Q1. Please improve English writing.

Reply: Thank you very much for your comment. The manuscript has been revised and corrected.

Q2. Electrochemical investigations are incomplete and have some errors:

Electrochemical behavior investigation can be completed more by calculating the diffusion coefficient or alpha. The data in Figure 2 do not match the data in Table 2.

Reply: Thank you very much for your question and comment. Your question made the manuscript more interesting to read. We calculated the diffusion coefficients and added the data to Table 2. The paper also compares and discusses performance with isolobal counterparts. The data for picture 2 was verified with the table.

Q3 Line 160: The meaning of the sentence “The color change is controlled by the difference between forward and reverse current.” is unclear. Voltammetric currents are in the range of microamperes and do not lead to color change!

Reply: Thank you for your comment. In this case, preparative electrolysis was carried out, where 1 F of electricity was passed. We have added an appropriate explanation to the text.

Q4 The reference of the article data used in Table 2 should be specified in the table.

Reply: Thank you for your comment. The references have been added.

Q6 The electrochemical behavior of the Ferrocene/Ferrocenium couple is completely reversible, that is, the peak current depends on the total concentration of these two, not on the concentration of each separately. As a result, the argument presented for Figure 5 is completely wrong (Lines 190 and 191).

Reply: Thank you for your question. We sincerely apologize that we cannot agree with you. In the case of ascorbic acid additions in the absence of ferrocene, the oxidation potential is shifted 0.12V more positively. In Figure 5, we give an example where the concentration of ferrocene does not change, but only ascorbic acid. For a better understanding of our interpretation, we also added the CVs of pure ascorbic acid.

Q7 Line 136: Scheme 6 is referenced, while there is no Scheme 6 in the main article and supplementary Information file.

Reply: Thank you for finding the typo. It's been corrected.

Q8 The caption of scheme 4 is not complete, write it in a complete form.

Reply: The caption of scheme has been corrected.

Q9 The title of table 2 is wrong, please correct it.

Reply: Thank you very much for your comment. The title of table has been corrected.

Reviewer 2 Report

The authors dedigned and synthized a seres of ionic liquids (ILs) with Ferrocene functional group on the IL cations, and detected their electrochemical  performance and applications in sensors. This is avery good work for IL and may have more prospective applications in electrochemical devices. The paper is well organized and the experimental results support their conclusions well. The reviewer remmends acceptance of this manuscript. 

Author Response

Dear Reviewer,

Herewith we submit a revised manuscript by Vadim V. Ermolaev, Liliya R. Kadyrgulova, Mikhail N. Khrizanforov*, Tatiana P. Gerasimova, Gulnaz R. Baembitova, Anna A. Lazareva, Vasili A. Miluykov «Conductive mediators in oxidation based on ferrocene func-tionalized phosphonium ionic liquids» to the International Journal of Molecular Sciences for participation in the thematic issue « Recent Advances in Novel Compositions for Electrochemical Applications)»

First of all, we would like to thank the Reviewers for inspective reading and reviewing of our manuscript and their valuable remarks.

We are confident that our results provide new important insights into the organic, organophosphorus and organometallic chemistry and such would be of a substantial interest to the broad interdisciplinary readership of the “International Journal of Molecular Sciences”. We therefore strongly hope that our work can be accepted for publication in your Journal.

We also provide the detailed answers to the reviewers’ remarks.

Please, do not hesitate to contact me if you have any problems or questions regarding our manuscript or if you have difficulty opening the files.

Yours sincerely,

Dr. Mikhail Khrizanforov,

======================================

Response to Reviewer 2 Comments

The authors dedigned and synthized a seres of ionic liquids (ILs) with Ferrocene functional group on the IL cations, and detected their electrochemical  performance and applications in sensors. This is avery good work for IL and may have more prospective applications in electrochemical devices. The paper is well organized and the experimental results support their conclusions well. The reviewer remmends acceptance of this manuscript. 

Reply: We are very grateful to you for your appreciation of the work.

Reviewer 3 Report

In this paper, the researchers described the functionalization of ferocene compounds with different acyl groups using the classical Friedel-Krafts method, followed by the quaternization of tBu3P to form phosphonium compounds. These compounds were fully characterised by spectroscopic methods and TG-DSC

By cyclic volammetry, the properties of the synthesised phosphoniums showed that the replacement of n-Bu moieties (studies described in the literature) at the phosphorus atom by tBu leads to a more anodic potential shift. The CV curves of Fc-C(O)(CH2)nP+(tBu)3X- and Fc-(CH2)n+1P+(tBu)3X- do not show a large difference between the forward and reverse currents. The authors demonstrate that the replacement of the C=O group by the CH2 moiety has a significant effect on the electrochemical properties of ferrocene salts, whose oxidation potential is close to that of pure ferrocene.

This article is well written and the reactions and analysis are good, but there are some errors, the authors should check the article.

- for example: page 2 line 46 "encanced properties" the authors mean "enhanced" ?

- Page 4 lines 124, 125 "Phosphonium bis(trifluoromethanesulfonyl)imide salts (5a-d) was obtained from bromides 3a-d by anion exchange reaction with excess of lithium bis(trifluoromethanesulfonyl)imide in water solution.". the author should delete this sentence.

- I suggest that the authors add the missing 13C for some molecules.

- I suggest that the authors add the IR spectra of the phosphonium compounds.

- In my opinion, there are similarities between the abstract and the conclusions, the authors should modify the conclusions.

Author Response

Dear Reviewer,

Herewith we submit a revised manuscript by Vadim V. Ermolaev, Liliya R. Kadyrgulova, Mikhail N. Khrizanforov*, Tatiana P. Gerasimova, Gulnaz R. Baembitova, Anna A. Lazareva, Vasili A. Miluykov «Conductive mediators in oxidation based on ferrocene func-tionalized phosphonium ionic liquids» to the International Journal of Molecular Sciences for participation in the thematic issue « Recent Advances in Novel Compositions for Electrochemical Applications)»

First of all, we would like to thank the Reviewers for inspective reading and reviewing of our manuscript and their valuable remarks.

We are confident that our results provide new important insights into the organic, organophosphorus and organometallic chemistry and such would be of a substantial interest to the broad interdisciplinary readership of the “International Journal of Molecular Sciences”. We therefore strongly hope that our work can be accepted for publication in your Journal.

We also provide the detailed answers to the reviewers’ remarks.

Please, do not hesitate to contact me if you have any problems or questions regarding our manuscript or if you have difficulty opening the files.

Yours sincerely,

Dr. Mikhail Khrizanforov,

======================================

Response to Reviewer 3 Comments

In this paper, the researchers described the functionalization of ferocene compounds with different acyl groups using the classical Friedel-Krafts method, followed by the quaternization of tBu3P to form phosphonium compounds. These compounds were fully characterised by spectroscopic methods and TG-DSC

By cyclic volammetry, the properties of the synthesised phosphoniums showed that the replacement of n-Bu moieties (studies described in the literature) at the phosphorus atom by tBu leads to a more anodic potential shift. The CV curves of Fc-C(O)(CH2)nP+(tBu)3X- and Fc-(CH2)n+1P+(tBu)3X- do not show a large difference between the forward and reverse currents. The authors demonstrate that the replacement of the C=O group by the CH2 moiety has a significant effect on the electrochemical properties of ferrocene salts, whose oxidation potential is close to that of pure ferrocene.

This article is well written and the reactions and analysis are good, but there are some errors, the authors should check the article.

Reply: We are very grateful to you for your appreciation of the work.

Q1. - for example: page 2 line 46 "encanced properties" the authors mean "enhanced" ?
Reply: Thank you for finding the typo. It's been changed.

Q2. - Page 4 lines 124, 125 "Phosphonium bis(trifluoromethanesulfonyl)imide salts (5a-d) was obtained from bromides 3a-d by anion exchange reaction with excess of lithium bis(trifluoromethanesulfonyl)imide in water solution.". the author should delete this sentence.

Reply: Thank you for the comment. It's been removed.

Q3 - I suggest that the authors add the missing 13C for some molecules.
Reply: Thank you for the comment. It's been added.

Q4 - I suggest that the authors add the IR spectra of the phosphonium compounds.
Reply: Thank you for the comment. It's been added.

- In my opinion, there are similarities between the abstract and the conclusions, the authors should modify the conclusions.

Reply: Thank you for the comment. We changed the conclusion.

Round 2

Reviewer 1 Report

 Accept in present form